

# Differential expression of the glucose transporter gene *glcH* in response to glucose and light in marine picocyanobacteria

José Ángel Moreno-Cabezuelo, Antonio López-Lozano, Jesús Díez and José Manuel García-Fernández

Departamento de Bioquímica y Biología Molecular, Campus de Excelencia Internacional Agroalimentario CeiA3, Universidad de Córdoba, Córdoba, Spain

## ABSTRACT

**Background:** Our team discovered that *Prochlorococcus* can take up glucose, in a process that changes the transcriptional pattern of several genes involved in glucose metabolization. We have also shown that *glcH* encodes a very high affinity glucose transporter, and that glucose is taken up by natural *Prochlorococcus* populations. We demonstrated that the kinetic parameters of glucose uptake show significant diversity in different *Prochlorococcus* and *Synechococcus* strains. Here, we tested whether the transcriptional response of *glcH* to several glucose concentrations and light conditions was also different depending on the studied strain.

**Methods:** Cultures were grown in the light, supplemented with five different glucose concentrations or subjected to darkness, and cells harvested after 24 h of treatment. qRT-PCR was used to determine *glcH* expression in four *Prochlorococcus* and two *Synechococcus* strains.

**Results:** In all studied strains *glcH* was expressed in the absence of glucose, and it increased upon glucose addition to cultures. The changes differed depending on the strain, both in the magnitude and in the way cells responded to the tested glucose concentrations. Unlike the other strains, *Synechococcus* BL107 showed the maximum glucose uptake at 5 nM glucose. Darkness induced a strong decrease in *glcH* expression, especially remarkable in *Prochlorococcus* MIT9313.

**Discussion:** Our results suggest that marine picocyanobacteria are actively monitoring the availability of glucose, to upregulate *glcH* expression in order to exploit the presence of sugars in the environment. The diverse responses observed in different strains suggest that the transcriptional regulation of glucose uptake has been adjusted by evolutive selection. Darkness promotes a strong decrease in *glcH* expression in all studied strains, which fits with previous results on glucose uptake in *Prochlorococcus*. Overall, this work reinforces the importance of mixotrophy for marine picocyanobacteria.

Corresponding author
José Manuel García-Fernández,
jmgarcia@uco.es

## INTRODUCTION

Cyanobacteria are organisms of ancient origin that perform oxygenic photosynthesis (*Stanier & Cohen-Bazire, 1977*), fixing $CO_2$ from the atmosphere to produce organic compounds, while releasing oxygen in the process (*Blankenship & Hartman, 1998*). The development of the photosynthetic apparatus, which was later transferred to plants by endosymbiosis, makes them responsible for the presence of oxygen in the atmosphere of Earth. Thus, cyanobacterial metabolism has profound implications at the global scale, from early in the history of our planet, to the present times, when their ubiquity and abundance confers them a huge influence in biogeochemistry and ecology (*Braakman, 2018*; *Braakman, Follows & Chisholm, 2017*).

Cyanobacteria have a great ability to adapt to different environments: from deserts (*García-Pichel & Pringault, 2001*) to the deep continental subsurface (*Puente-Sánchez et al., 2018*), from temperate oceans (*Hoffman, 1999*) to polar waters (*Vincent, 2000*). Some of the key features explaining this surprising adaptation potential is their genetic diversity and their capability to exploit different resources, from atmospheric dinitrogen to organic compounds where available (*Beck et al., 2012*; *Biller et al., 2014*; *Dufresne et al., 2008*; *Rocap et al., 2003*; *Scanlan et al., 2009*; *Shih et al., 2013*). In marine environments, the distribution of marine picocyanobacterial ecotypes has been extensively studied, showing a clear partition for the *Prochlorococcus* clades, but overlapping physiologies and environmental distributions for marine *Synechococcus* (*Farrant et al., 2016*; *Kent et al., 2018*).

The utilization of organic compounds (including sugars) is well documented in many, but not all, genera of cyanobacteria (*Pelroy, Rippka & Stanier, 1972*; *Picossi, Flores & Ekman, 2013*; *Rippka et al., 1979*). While photosynthesis allows the biosynthesis of sugars from $CO_2$, the uptake of sugars when available is beneficial in terms of bioenergetics (*Muñoz-Marín et al., 2013*). Therefore, some cyanobacterial strains have developed systems to take up sugars from the environment. The gene *glcP* was identified to encode a fructose-glucose permease in *Synechocystis* sp. PCC 6803 (*Schmetterer, 1990*; *Zhang et al., 1989*). Interestingly, a GlcP permease was demonstrated to be necessary for *Nostoc punctiforme* to form symbiosis with the plant *Anthoceros punctatus* (*Ekman et al., 2013*); furthermore, multiple ABC sugar transporters have been shown in *Anabaena* sp. PCC 7120 (*Nieves-Morión & Flores, 2018*). Recent studies have suggested diverse roles for GlcP transporters in free-living and symbiotic cyanobacteria (*Picossi, Flores & Ekman, 2013*).

Despite the fact that glucose transporters had been described long ago in several genera of cyanobacteria, at the beginning of this century it was widely considered that marine picocyanobacteria were not capable of using glucose. The small size of their genomes (*Dufresne et al., 2003*; *Palenik et al., 2003*; *Rocap et al., 2003*) and the very scarce concentration of sugars in the oceans (in the nanomolar range; *Muñoz-Marín et al., 2013*; *Zubkov et al., 2008*) suggested these organisms were restricted to producing glucose from photosynthesis. Furthermore, no glucose transporter had been identified in the genomes of marine picocyanobacteria. Our team discovered that glucose was taken up by

*Prochlorococcus*, with a clear effect on the expression of different related genes (*Gómez-Baena et al., 2008*). In addition, we identified the gene Pro1404/*glcH* to encode a previously unknown, very high affinity glucose transporter in *Prochlorococcus* sp. strain SS120 (*Muñoz-Marín et al., 2013*), and demonstrated glucose uptake by natural *Prochlorococcus* populations. This transporter was shown to have a significant diversity in the transport kinetics (*Muñoz-Marín et al., 2017*). Besides, the phylogeny of *glcH* is very similar to the consensus phylogeny of marine picocyanobacteria, suggesting this gene has been subjected to evolutive selection in the different clades of *Prochlorococcus* and *Synechococcus*. This is in good agreement with metagenomic studies which have shown a large capacity for mixotrophy in marine picocyanobacteria (*Yelton et al., 2016*), confirmed by field studies demonstrating the uptake of organic compounds by these microorganisms (*Björkman et al., 2015*; *Duhamel, Björkman & Karl, 2012*; *Duhamel et al., 2018*; *Michelou, Cottrell & Kirchman, 2007*; *Talarmin et al., 2011*; *Vila-Costa et al., 2006*; *Zubkov et al., 2003*). Moreover, field studies also suggested that the growth of natural *Prochlorococcus* populations was enhanced after glucose addition (*Moisander et al., 2012*).

All these studies suggest that *glcH* encodes a transporter with an important role for the picocyanobacterial populations in the oceans, whose biological function is being fine-tuned by evolutive selection. If this hypothesis holds true, it would be expectable to find different responses regarding the glucose uptake in the different *Prochlorococcus* and *Synechococcus* ecotypes, reflecting their adaptation to different niches. To test this hypothesis, we decided to study the regulation of *glcH* expression in six marine cyanobacterial strains, in laboratory cultures subjected to different glucose concentrations.

Darkness has been shown to induce a decrease in the glucose uptake levels in *Prochlorococcus* sp. SS120 (*Gómez-Baena et al., 2008*). Furthermore, addition of the inhibitors of the photosynthetic electron transport DCMU and DBMIB also inhibited glucose uptake in the same strain (*Muñoz-Marín et al., 2017*). Therefore, we decided to study the effect of darkness on the expression of *glcH*, to assess the possible diversity of responses in different cyanobacterial strains.

Our results show a diversity of responses depending on the strain, indicating that *glcH* expression is regulated differently depending on the studied ecotype. This reinforces the importance of glucose uptake for the ecology of *Prochlorococcus* and *Synechococcus*, and sheds light on the relevance of mixotrophy for these ecologically important microorganisms.

## MATERIALS AND METHODS

### Growth of *Prochlorococcus* and *Synechococcus* cultures

We used six model strains of marine picocyanobacteria: four *Prochlorococcus* and two *Synechococcus*. For *Prochlorococcus*, we studied two low-light strains, SS120 (*Chisholm et al., 1992*; *Dufresne et al., 2003*) and MIT9313 (*Rocap et al., 2003*), which differed in their evolutive origins (late- and early-branching, from the clades LL IV and LL II/III, respectively); and two high-light strains, PCC 9511 (*Rippka et al., 2000*; *Rocap et al., 2003*) and TAK9803-2 (*Garczarek et al., 2000*) (from the clades HLI and HLII,

respectively). These strains were chosen because they are representative of the main adaptations in *Prochlorococcus* (high light vs low light) and also of some of the main clades described in this cyanobacterium.

The phylogenetic diversity within the *Synechococcus* genus is larger than in *Prochlorococcus*, and addressing *glcH* expression in *Synechococcus* strains of every subclade was out of the scope of this study. Hence we chose two model strains for comparison with *Prochlorococcus*, given they coexist in many marine environments: BL107 (abundant in cold coastal waters (*Dufresne et al., 2008*; *Six et al., 2007*)) and WH7803 (*Dufresne et al., 2008*; *Kursar, Swift & Alberte, 1981*) (unclear distribution), corresponding to clades IV and V, respectively.

*Prochlorococcus marinus* sp. strains PCC 9511, TAK9803-2, SS120 and MIT9313 were routinely cultured in polycarbonate flasks (*Nalgene)* using PCR-S11 medium as described earlier (*El Alaoui et al., 2001*). The seawater used as basis for this medium was obtained from the Mediterranean Sea, near Málaga, kindly provided by the Instituto Español de Oceanografía (Spain). Cultures were grown in a culture room at 24 °C under continuous blue irradiance (4 or 40 µE m$^{-2}$ s$^{-1}$, for low- and high-light adapted strains, respectively). Growth was determined by measuring the absorbance of cultures at 674 nm and cells were collected during the exponential phase of growth ($A_{674}$ = 0.05).

*Synechococcus* sp. strains WH7803 and BL107 were grown in a chemically defined artificial seawater medium (*Moore et al., 2007*). Cells were grown in polycarbonate Nalgene flasks, in a culture room under continuous blue light at 40 µE m$^{-2}$ s$^{-1}$ and 24 °C. Growth was determined by measuring the absorbance of cultures at 550 nm (strain WH7803) or 495 nm (strain BL107) and cells were collected during the exponential phase of growth, when absorbance at the indicated wavelengths was 0.1.

## Cell collection

*Prochlorococcus* cultures reaching 0.05 units of absorbance at 674 nm were split into several aliquots: one used as control culture, while glucose (at the indicated concentrations) was added to the others. The same strategy was followed with *Synechococcus* cultures, with the exception that the starting cultures reached 0.1 units of absorbance at 550 or 495 nm. They were then kept under standard light and temperature conditions and the cells were collected at the indicated times. In experiments addressing the effect of darkness, we divided cyanobacterial cultures of 1 L in two aliquots: one of them was kept under standard irradiance (control) and the other one was subjected to darkness by covering the bottles with two layers of aluminium foil. After 24 h in these conditions, cells were harvested and used to determine *glcH* expression. Sampling of cultures under darkness was carried out with as low light as possible. Cells were harvested at 26,000 *g* for 8 min at 4 °C using an *Avanti J-25 Beckman* centrifuge equipped with a JA-14 rotor. After pouring most of the supernatant and carefully pipetting out the remaining medium, the pellet was directly resuspended in 10 mM sodium acetate (pH 4.5) supplemented with 200 mM sucrose and 5 mM EDTA. Samples were stored at −80 °C until used.
## RNA isolation

RNA was isolated from 500 mL culture samples, using the *TRIsure RNA Isolation Reagent* (Bioline, London, United Kingdom) following the method described previously (*Domínguez-Martín, Díez & García-Fernández, 2016*).

## Real-time quantitative RT-PCR analysis of gene expression

By using specific primers to amplify *glcH* and *rnpB* (which was chosen as the housekeeping gene for standardization), we studied changes in *glcH* expression by qRT-PCR in cultures of the above described strains subjected to different conditions. In order to study the effect of glucose availability on the expression of the *glcH* gene, we selected a range of glucose concentrations representative of the conditions found in oligotrophic oceans (where this sugar is found in nanomolar levels (*Muñoz-Marín et al., 2013*)), but included also two higher concentrations to test whether they induced a significant change, as observed previously in other studies (*Coe et al., 2016*). Therefore, we used cultures with 0 (control), 1, 5, 100 and 1,000 nM glucose. Glucose was added to samples at zero time, and cells were harvested after 24 h in order to check for sustained metabolic responses, given the slow growth of *Prochlorococcus* (*Vaulot et al., 1995*).

cDNA synthesis from RNA samples was carried out with the *qScript^{TM} cDNA Synthesis* kit from QuantaBio, Beverly, Mass., USA as recommended by the manufacturer. Specific primers for the genes *glcH* (encoding a high affinity glucose transporter, described previously; *Muñoz-Marín et al., 2017*, *2013*) and *rnpB* (encoding RNase P and used to standardize *glcH* expression) were designed by using the software *Primer3 Plus* (http://www.bioinformatics.nl/cgi-bin/primer3plus/primer3plus.cgi), on the base of the published genomes of the studied cyanobacterial strains.

In the case of *Prochlorococcus* sp. strain TAK9803-2, for which no genomic sequence is available, we used primers designed to amplify a 189 nucleotides fragment of the *glcH* gene (previously annotated as Pro1404/*melB* in *Prochlorococcus* sp. strain SS120) in the strains MED4 and MIT9515. The sequences of the primers used were:

forward, 5′-ATGCTCTCTTATGGACTTGGAG-3′;
reverse, 5′-TTTTGTCCTATCACTTAACCATCC-3′.

When these primers were used in a PCR using genomic DNA of *Prochlorococcus* sp. strain TAK9803-2, we obtained two DNA fragments. One of them (sequence available in Supplementary Material) was sequenced and confirmed to belong to the *glcH* gene. This sequence was used to design specific primers for performing qRT-PCR with samples of *Prochlorococcus* sp. strain TAK9803-2 (Supplemental File 1).

Specificity of the qRT-PCR reactions was checked by agarose gel electrophoresis. The sequences of the primers used are listed in Table 1.

qRT-PCR semi-quantitative gene expression determinations were carried out as described (*Domínguez-Martín et al., 2014*), according to the Pfaffl method (*Pfaffl, 2001*), on an *iCycler* IQ multicolor real time PCR detection system (Bio-Rad, Hercules, California, USA). The melting point of PCR products was used to confirm the absence of false

**Table 1 Primers used in qRT-PCR reactions to determine gene expression.**

| Cyanobacterial strain | | glcH | rnpB |
|---|---|---|---|
| *Prochlorococcus* sp. SS120 | Forward | 5′-GCTTTTATGGCAGGTTCTTT-3′ | 5′-CTCTCGGTTGAGGAAAGTC-3′ |
| | Reverse | 5′-CAAATAGCCGCAAGACTCAG-3′ | 5′-CCTTGCCTGTGCTCTATG-3′ |
| *Prochlorococcus* sp. MIT9313 | Forward | 5′-GGGCTTTACCTGTTGCTCTG-3′ | 5′-AAGACGAGCTTGGTTGAGGA-3′ |
| | Reverse | 5′-CAAGCAGCGATCCATAGACA-3′ | 5′-CTCTTACCGCACCTTTGCAC-3′ |
| *Prochlorococcus* sp. TAK9803-2 | Forward | 5′-ACTGCATCCCATATCTTTATTAA-3′ | 5′-ACAGAAACATACCGCCTAAT-3′ |
| | Reverse | 5′-ACGCAATTTGGTTTTTTTCT-3′ | 5′-ACCTAGCCAACACTTCTCAA-3′ |
| *Prochlorococcus* sp. PCC 9511 | Forward | 5′-GTCTAGCCGCCACACAATTT-3′ | 5′-ACAGAAACATACCGCCTAAT-3′ |
| | Reverse | 5′-TGCAGCAATTAGCATCCAAG-3′ | 5′-ACCTAGCCAACACTTCTCAA-3′ |
| *Synechococcus* sp. BL107 | Forward | 5′-ATCATGCTGGTGGGTCTAGG-3′ | 5′-CAAGGCCAAGGAACGATG-3′ |
| | Reverse | 5′-GTGTACAGCCCGGCAGGT-3′ | 5′-GCAGAGGGTGGGTGGTTAT-3′ |
| *Synechococcus* sp. WH7803 | Forward | 5′-CTATACCGCCTGGATGGTGT-3′ | 5′-TGAGGAGAGTGCCACAGAAA-3′ |
| | Reverse | 5′-AATCAGACCCATGCAGATCC-3′ | 5′-GTTTACCGAGCCAGCACCT-3′ |

amplifications. Triplicate determinations from at least three independent biological samples subjected to identical culture conditions were used for calculations. Results were endogenously normalized to *rnpB* expression using the $2^{-\Delta\Delta Ct}$ method (*Pfaffl, 2001*), since the expression of this gene did not change under our experimental conditions.

The results shown in Figs. 1–4 correspond to relative gene expression, so that each value represents the changes under a specific condition with respect to the same culture under control conditions. Raw results are provided in Supplemental File 2.

### *Prochlorococcus* and *Synechococcus* genomic sequences

Genomic sequences were retrieved from the Joint Genome Institute (http://genome.jgi-psf.org) and Cyorf (http://cyano.genome.ad.jp/) databases.

### Statistical analysis

Results were obtained by using at least three different samples, and error bars show the standard deviation. Statistical analysis was done using the Student's unpaired *t*-test, by using the software *GraphPad Prism 7 for Mac OS X* (GraphPad Software, San Diego, California, USA). Assessment of data significance is indicated with asterisks: *indicates $p \leq 0.05$; **indicates $p \leq 0.01$; ***indicates $p = 0.0001$; ****indicates $p < 0.0001$. For all values, significance is expressed with respect to the results of the control value (culture with no glucose addition for Figs. 1 and 2, and culture grown under light for Figs. 3 and 4).

## RESULTS

### Effect of different glucose concentrations on *glcH* expression

Figure 1A shows the results obtained with one of the most studied *Prochlorococcus* low-light strains, SS120. We observed a weak increase (1.5–2.5 fold) in *glcH* expression, in samples subjected up to 100 nM glucose, but there was a sharp increase (more than 6-fold) in cultures where 1,000 nM glucose had been added. The increases were significant for 1 nM ($p = 0.0295$) and 1,000 nM glucose ($p = 0.0016$).

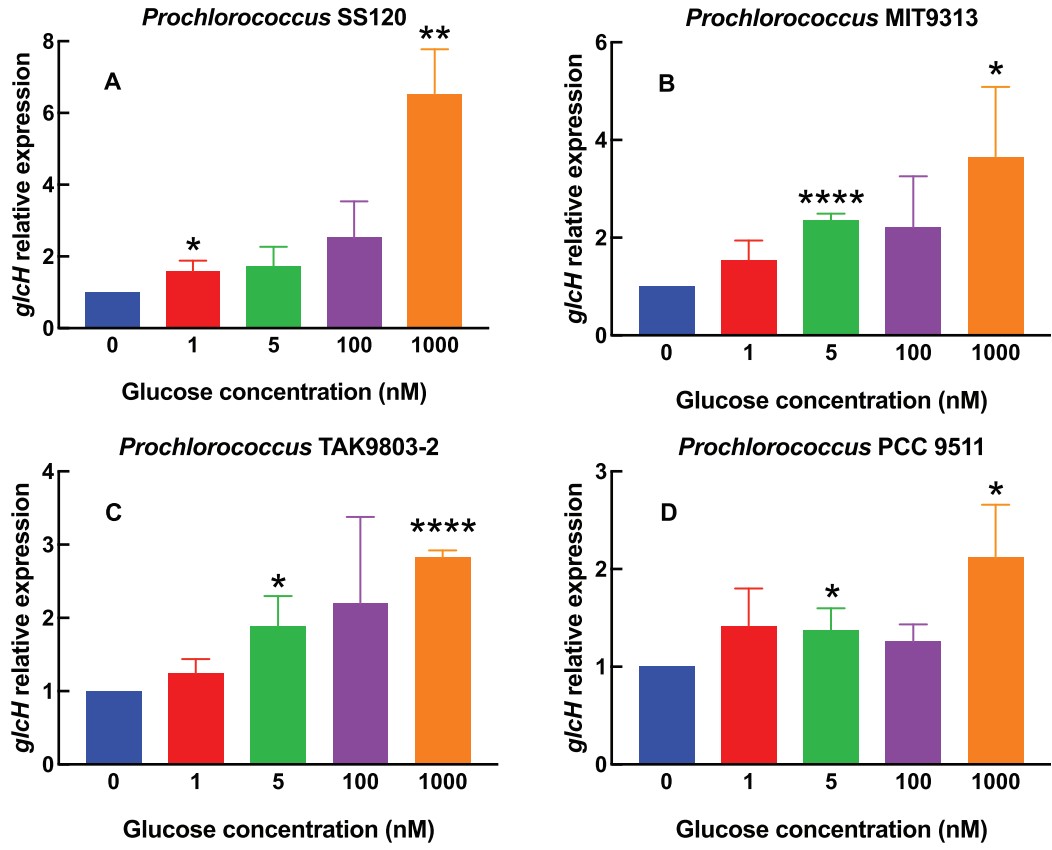

**Figure 1 Effect of different glucose concentrations on *glcH* expression in *Prochlorococcus* strains.** The indicated glucose concentrations were added at zero time to cultures of *Prochlorococcus* sp. strains SS120 (A), MIT9313 (B), TAK9803-2 (C) and PCC 9511 (D). Cells were harvested after 24 h. *glcH* expression was determined by qRT-PCR. Error bars show the standard deviation and asterisks indicate the statistical significance with respect to the control cultures (no glucose addition).

The response in *Prochlorococcus* MIT9313 (Fig. 1B) was more progressive, showing an almost linear rise in *glcH* expression along the range of studied glucose concentrations, with the only exception of cultures with 100 nM glucose. The value of *glcH* expression observed at 5 nM glucose was highly significant ($p < 0.0001$). However, there was no significant difference when comparing the response of strains SS120 vs MIT9313 to 1,000 nM glucose ($p = 0.8628$).

*glcH* expression also increased progressively in *Prochlorococcus* sp. strain TAK9803-2 (Fig. 1C), but in a lower proportion when compared to the previously described strains: in this case, the maximum values (determined also at 1,000 nM glucose) averaged less than a 3-fold increase, which is ca. half of the values determined for *Prochlorococcus* SS120. Increases were significant at 5 nM ($p = 0.0225$) and highly significant at 1,000 nM ($p < 0.0001$).

For *Prochlorococcus* PCC 9511 (Fig. 1D), the glucose concentrations of 1, 5 and 100 nM had little effect (a marginal increase, with an average value below 1.5-fold with respect to the control culture). The only clear increase was observed at the highest glucose

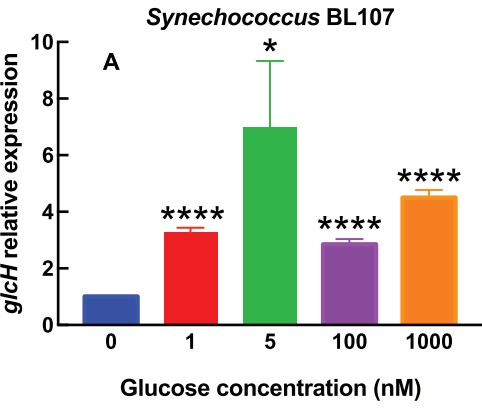
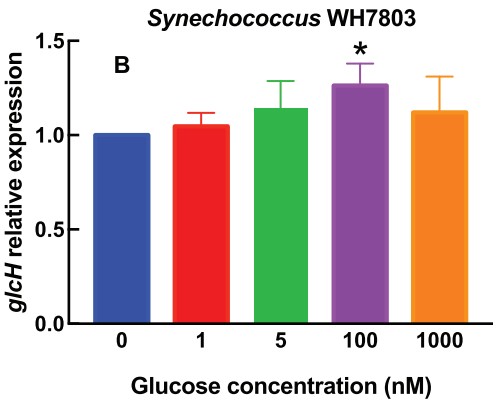

**Figure 2 Effect of different glucose concentrations on *glcH* expression in marine *Synechococcus* strains.** Experimental conditions are the same as in Fig. 1, but using *Synechococcus* sp. strains BL107 (A) and WH7803 (B).

concentration, where *glcH* expression was 2-fold higher. However, increases were significant at 5 and 1,000 nM glucose ($p$ = 0.0453 and 0.0241, respectively). The maximum value found for HL strains was lower than that observed in LL *Prochlorococcus* strains (i.e., 3.65 for MIT9313 and 6.52 for SS120, vs 2.82 for TAK9803-2 and 2.11 for PCC 9511).

In *Synechococcus* sp. BL107 (Fig. 2A), the response was clearly different with respect to all tested *Prochlorococcus* strains: *glcH* expression showed a peak at 5 nM, with a 7-fold increase, while the other tested concentrations induced lower increases. It is worth noting that this is the only studied cyanobacterial strain where the maximum *glcH* expression was not observed at the maximum tested glucose concentration. Besides, all observed increases were significant (5 nM glucose, $p$ = 0.0115) or highly significant (1, 100 and 1,000 nM glucose, $p$ < 0.0001).

By contrast, *Synechococcus* sp. WH7803 does respond to glucose addition in a progressive manner (Fig. 2B), unlike strain BL107, although the overall effect is minor, with very weak increases at all tested concentrations, with the only exception of 1,000 nM glucose. In this case, only the value observed at 100 nM was significant ($p$ = 0.0170).

When comparing the results for the six tested cyanobacterial strains, it is remarkable that the response in *Synechococcus* clearly differs from *Prochlorococcus*, and exhibits two almost antagonistic profiles: from the almost no-change of WH7803, to the bell shape observed in BL107. This suggests that a more detailed study within the *Synechococcus* genus would probably disclose a remarkable diversity in the response of *glcH* expression, in good agreement with the reported phylogenic diversity of this genus.

## Effect of light on *glcH* expression

Figure 3 shows the results obtained in *Prochlorococcus*, where *glcH* expression determined in control cultures for each strain (growing under light) received the value of 1. Thus the first bar in Fig. 3 serves for comparison to all strains shown at the right part of the chart. Darkness provoked a strong decrease in *glcH* expression in all strains. This decrease ranged from a minimum of a significant 60% (in the case of *Prochlorococcus* sp. SS120; $p$ = 0.0035) to an almost complete inhibition of *glcH* expression (97% decrease in

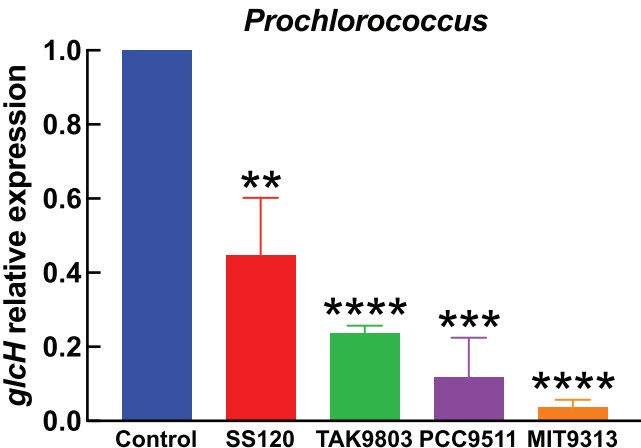

**Figure 3 Effect of darkness on *glcH* expression in *Prochlorococcus*.** Cultures of the indicated *Prochlorococcus* strains were divided in two aliquots at zero time. One of them was kept under standard conditions, the other one was subjected to darkness. Cells were harvested after 24 h. *glcH* expression was determined by qRT-PCR. Error bars show the standard deviation, and asterisks indicate the statistical significance with respect to the control cultures (growing under light).

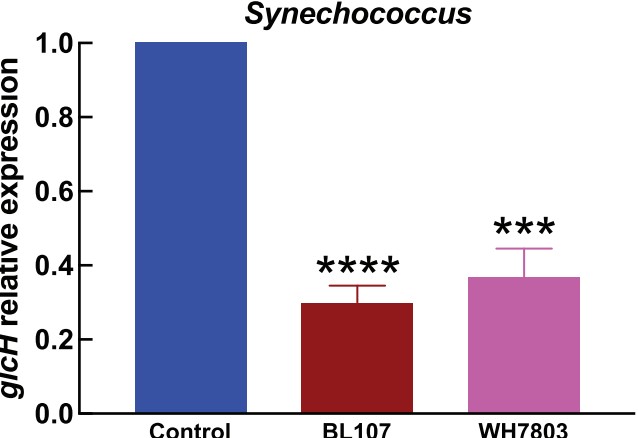

**Figure 4 Effect of darkness on *glcH* expression in *Synechococcus*.** Experimental conditions are the same as in Fig. 3.

the case of *Prochlorococcus* sp. strain MIT9313, very significant; $p < 0.0001$). These results highlight the lack of similarity in the responses of the two studied low-light adapted strain. We found an intermediate situation in the strains TAK9803-2 and PCC 9511, with darkness-promoted decreases of 76% (very significant, $p < 0.0001$) and 88% in *glcH* expression (very significant, $p = 0.0001$), respectively.

In the experiments studying the effect of darkness in *Synechococcus* strains (Fig. 4), we observed also a strong effect on *glcH* expression, which was reduced in a 70% in the case of strain BL107 (very significant, $p < 0.0001$), and in a 63% in the case of strain WH7803 (very significant, $p = 0.0001$). These figures are similar to those observed for *Prochlorococcus* sp. SS120, and lower compared to those for the other *Prochlorococcus* strains.

## DISCUSSION

The utilization of sugars by some cyanobacterial strains was reported long ago, but the molecular mechanisms allowing sugar uptake is a process less known. Initial studies demonstrated the presence of a glucose transporter due to its ability to transport fructose as well, which is toxic for some strains (as *Synechocystis* sp. PCC 6803 and PCC 6714): mutant strains resistant to fructose allowed to show the ability to take up glucose (*Flores & Schmetterer, 1986*). Physiological characterization of glucose uptake in *Synechocystis* showed the process had a $K_s$ constant in the millimolar range (*Joset et al., 1988*). $K_s$ is the glucose concentration at which glucose transport reaches half of its maximum value, for a given transporter. Complementation studies identified the gene *glcP*, encoding a sugar/$H^+$ symporter, as the gene encoding the glucose transporter in *Synechocystis* sp. PCC 6803 (*Zhang et al., 1989*). This gene is homologous to other glucose transporters found in bacteria and animals (*Zhang et al., 1989*). A different type of glucose transporter was discovered in *Prochlorococcus* (*Muñoz-Marín et al., 2013*), initially annotated as *melB* due to its homology with melibiose transporters described in *E. coli* (*Hanatani et al., 1984*).

An intriguing fact is that *glcH* is present in all sequenced cyanobacterial genomes, but no specific function had been assigned to it thus far. This ubiquity suggests it encodes an important protein for cyanobacteria, which is in contrast with the previous discovery of GlcP as a low affinity glucose transporter: if *glcH* is only required in environments with very low glucose concentrations (such as oligotrophic oceans), it is difficult to understand why it has been conserved in all sequenced cyanobacteria thus far. Previous reports showed that the phylogeny of *glcH* mirrors the consensus phylogeny of marine picocyanobacteria (*Muñoz-Marín et al., 2017*) and a significant diversity in *Prochlorococcus* glucose uptake (*Gómez-Baena et al., 2008*; *Muñoz-Marín et al., 2017*). All those results suggest that *glcH* has been subjected to selective evolution in marine picocyanobacteria, where its physiological function is being modulated to better fit the ecological niche where they thrive. On this basis, we hypothesized that the regulation of glucose uptake (and more specifically, of the expression of the gene encoding the glucose transporter) would show a certain level of diversity as well, reflecting the different habitats where model picocyanobacterial strains were isolated. This prompted us to study the *glcH* expression in four *Prochlorococcus* and two *Synechococcus* strains, subjected to changes in two key parameters representative of their environmental conditions: availability of different glucose concentrations, and the effect of darkness.

The results presented here confirm our hypothesis: we observed differential *glcH* expression under the studied conditions, both in the magnitude of the observed changes and in the ways these changes happened.

Regarding the magnitude of changes, we could find maximum increases of ca. 7-fold in the *glcH* expression for *Prochlorococcus* sp. strain SS120 (low-light adapted) and *Synechococcus* sp. strain BL107 (see Figs. 1A and 2A), vs minimum increases of ca. 2-fold for *Prochlorococcus* TAK9803-2 and PCC 9511 (high-light adapted; Figs. 1C and 1D), and even lower in *Synechococcus* sp. WH7803 (Fig. 2B).

It is noteworthy that *Prochlorococcus* sp. strains SS120 and MIT9313 showed a higher increase in *glcH* expression (3.5–8 fold), when compared to strains TAK9803-2 and PCC 9511 (1.6–2.9 fold). These results are in agreement with previous reports showing higher glucose uptake levels also in low-light adapted *Prochlorococcus* strains, such as SS120 or MIT9303 (*Muñoz-Marín et al., 2017*). This might suggest that low-light adapted strains hold a higher capacity to upregulate *glcH* expression than high-light adapted ecotypes. This hypothesis would require to analyze a larger number of strains for confirmation. If this is true, it would fit to the idea that glucose utilization is more useful for marine picocyanobacterial strains adapted to life with very little light available (*Jiao et al., 2013*), where glucose could support metabolism and allow survival under extended darkness (*Coe et al., 2016*).

Interestingly, the maximum value was observed at 1,000 nM glucose in SS120, but not in BL107, which had maximum glucose uptake at 5 nM glucose. This might suggest that the regulation of *glcH* expression has been modified, so that *Synechococcus* sp. strain BL107 is more sensitive to variations in the nanomolar range of glucose concentrations, which might happen often in its habitat: this is a coastal strain which was isolated from deep samples, at the Blanes Bay (*Dufresne et al., 2008*; *Six et al., 2007*); while all other studied strains (from both genera, *Prochlorococcus* and *Synechococcus*) show the maximum value at the highest tested glucose concentration. Additional information regarding sugar concentrations at the deep waters where *Synechococcus* sp. BL107 was isolated would be required for confirmation.

Since transport and consumption of glucose is beneficial for marine picocyanobacteria, in bioenergetic terms (with respect to glucose synthesis de novo (*Muñoz-Marín et al., 2013*)), it is advantageous for them to take up glucose from the environment when possible. Even if they need to use energy to take up glucose in the dark, still the balance is positive, as shown for instance in the extended survival under darkness which has been reported after glucose addition to *Prochlorococcus* cultures (*Coe et al., 2016*).

Expression levels can be grouped into three patterns: in some strains, there is a progressive increase (SS120, MIT9313, TAK9803-2); in others, there is little variation for most glucose concentrations (PCC 9511, WH7803); and finally, in the case of *Synechococcus* sp. BL107, the observed values have a peak at the mid tested concentration, unlike all other cyanobacterial strains here studied.

Our results confirm that *glcH* is expressed when no glucose was added to cultures, but its expression increased when glucose was added. This suggests that marine picocyanobacteria are monitoring the presence of glucose in the environment, so that they can upregulate glucose uptake to use it when available. The mechanism used by *Prochlorococcus* and *Synechococcus* to detect the presence of glucose in the environment is unknown. The sensor kinase Hik31 has been shown to be involved in glucose sensing in *Synechocystis* sp. strain PCC 6803 (*Kahlon et al., 2006*). However, this sensor kinase is missing in all known *Prochlorococcus* and marine *Synechococcus* genomes, and therefore the sensor molecule must be different in these microorganisms.

Glucose uptake was increased as well in *Plectonema boryanum* in the presence of glucose (*Raboy & Padan, 1978*). However, in other cyanobacterial strains the glucose

uptake is not inducible by glucose (*Beauclerk & Smith, 1978*; *Der-Vartanian, Joset-Espardellier & Astier, 1981*). It has to be mentioned that the mechanisms for glucose uptake previously discovered in cyanobacteria are of low affinity, and therefore it is expectable that their regulation and/or inducibility might be subjected to different rules than the very high affinity transporter encoded by *glcH* in marine picocyanobacteria. The large difference in the $K_s$ values for GlcH vs GlcP transporters (roughly 3,000 times (*Muñoz-Marín et al., 2013*) might explain why *glcH* expression is noticeably upregulated at nanomolar concentrations of glucose (i.e., Figs. 1 and 2), especially in the cases of *Prochlorococcus* sp. MIT9313 and *Synechococcus* sp. BL107. However, at such low glucose concentrations, probably the low affinity transporter is not being used by cyanobacterial strains harboring the *glcP* gene (although this has not been tested, to our knowledge). Since the function of *glcH* has not been addressed yet in non-marine strains, it would be possible that its function as a very high affinity glucose transporter has been overlooked thus far in those microorganisms. We are currently studying this topic in some model freshwater strains.

Darkness induced a strong decrease in *glcH* expression in all tested cyanobacterial strains (Figs. 3 and 4), which in some cases was almost completely arrested (i.e., *Prochlorococcus* sp. strain MIT9313). This was in good agreement with previous results showing that glucose uptake was ca. 42% inhibited by darkness in *Prochlorococcus* sp. strain SS120 (*Gómez-Baena et al., 2008*). Furthermore, addition of photosynthetic electron transport inhibitors also induced a remarkable decrease in glucose uptake: DCMU promoted a 50% decrease, while DBMIB abolished glucose uptake (*Muñoz-Marín et al., 2017*). All these results suggest that GlcH is an active transporter (*Muñoz-Marín et al., 2017*), and therefore picocyanobacterial cells are using metabolic energy in order to take up glucose, which is beneficial in energetic terms with respect to de novo biosynthesis (*Muñoz-Marín et al., 2013*). Consequently, the energy limitations imposed by darkness or photosynthetic inhibitors on the metabolism of marine picocyanobacteria explain the decrease in both glucose uptake and *glcH* expression.

The situation is however, different in freshwater strains using the GlcP transporter, which can take up glucose in darkness. For instance, *Aphanocapsa* sp. PCC 6714 was capable of glucose assimilation both in darkness or under light in the presence of 10 µM DCMU (*Pelroy, Rippka & Stanier, 1972*). Glucose uptake in *Aphanocapsa* sp. PCC 6714 was unaffected after cultures were subjected to darkness; however, preincubation in the dark for 48 h did induce an important decrease in glucose uptake (*Beauclerk & Smith, 1978*). Both *Aphanocapsa* sp. PCC 6714 and *Synechocystis* sp. PCC 6803 were shown to grow on glucose in the presence of 10 µM DCMU (*Der-Vartanian, Joset-Espardellier & Astier, 1981*; *Flores & Schmetterer, 1986*; *Schmetterer, 1990*), although *Synechocystis* sp. PCC 6803 showed extremely weak growth in the light with glucose with no DCMU (*Flores & Schmetterer, 1986*). *Plectonema boryanum* showed only a 20% decrease in glucose uptake after cells were subjected to darkness (*Raboy & Padan, 1978*). Growth on glucose, either under light or in darkness, has not been reported in any marine picocyanobacteria, to our knowledge. Rippka and coworkers however, reported that 10 mM glucose, fructose and sucrose were tolerated and seemed to

slightly prolong survival, but had no major beneficial effect on the growth of *Prochlorococcus marinus* PCC 9511 (*Rippka et al., 2000*). Nevertheless, field studies in the oligotrophic South Pacific Ocean suggested that glucose could support the growth of natural populations of *Prochlorococcus* (*Moisander et al., 2012*).

Light has been shown to stimulate the uptake of organic compounds in marine picocyanobacteria, including amino acids and ATP (*Duhamel, Björkman & Karl, 2012*; *Gómez-Pereira et al., 2013*; *Mary et al., 2008*; *Michelou, Cottrell & Kirchman, 2007*). Moreover, the work of Duhamel and coworkers with natural picocyanobacterial populations in the western tropical South Pacific Ocean fits very nicely with our results (*Duhamel et al., 2018*): they observed that light enhanced cell specific glucose uptake by ca. 50% for *Prochlorococcus* and *Synechococcus*. They proposed that variability in light availability (derived for instance from diel sunlight rhythms, cloud coverage, etc.,) could significantly impact glucose uptake in marine cyanobacteria. Given the similarity of their observations to the stimulation of *glcH* expression by light in *Prochlorococcus* and *Synechococcus* (Figs. 3 and 4), we propose that light availability controls glucose uptake in marine picocyanobacteria by transcriptional regulation of *glcH*.

Furthermore, studies using photosynthetic transport inhibitors suggest that incorporation of glucose into the cells can be partially supported by ATP produced by the cyclic electron transport around photosystem I. This would allow the maintenance of ca. 50% glucose uptake when photosystem II is not working, providing an important level of metabolic flexibility in marine picocyanobacteria (*Duhamel et al., 2018*; *Muñoz-Marín et al., 2017*).

In the limited set of cyanobacterial strains studied here, we found that darkness had a stronger effect on *glcH* expression in *Prochlorococcus* than in *Synechococcus* (Fig. 3 vs Fig. 4). If this situation is confirmed by further studies with additional strains, it might help to understand why *Synechococcus* seems to have a longer dark-survival capacity than *Prochlorococcus* (*Coe et al., 2016*): darkness would allow a higher glucose uptake level in *Synechococcus* than in *Prochlorococcus*, providing carbon and energy when photosynthesis is not possible, and therefore conferring an advantage to *Synechococcus* over *Prochlorococcus* when subjected to environmental conditions of very low to no light at all.

Overall, the results presented in this work suggest that *Prochlorococcus* and marine *Synechococcus* strains have modified the regulation of *glcH* transcription in their evolution. This is in good agreement with a recent metapangenomic paper, which revealed a small set of core genes occurring in hypervariable genomic islands of *Prochlorococcus* populations, all of them linked to sugar metabolism (*Delmont & Eren, 2018*). This study suggested that a high sequence diversity of sugar metabolism genes could confer benefits to *Prochlorococcus*. Such a high sequence diversity could be linked to the observed diversity in the transcriptional control mechanisms. Further work is required to understand the physiological relevance of sugar uptake and utilization for marine picocyanobacteria.

## CONCLUSIONS

Our results demonstrate that *Prochlorococcus* and marine *Synechococcus* express the gene *glcH* in the absence of added glucose and upregulate its expression upon addition of

increasing glucose concentrations. This suggests that marine picocyanobacteria are actively monitoring the environment for the presence of a valuable compound, such as glucose, and enhancing the glucose transporter transcription in order to fulfill some of their metabolic needs by sourcing sugars from the environment. The light enhancement observed for *glcH* expression mirrors that observed for glucose uptake in natural populations, suggesting that transcriptional regulation is the primary mechanism to regulate glucose uptake in marine picocyanobacteria.

The diversity observed in the response of *glcH* expression to the availability of glucose and light indicate that transcriptional regulation of *glcH* has been adjusted in the evolution of these microorganisms, to better fit the environment where they are the most abundant primary producers. The results here presented reinforce the importance of mixotrophy for marine cyanobacteria.

## ACKNOWLEDGEMENTS

We thank Dr. M.C. Muñoz-Marín and Dr. G. Gómez-Baena for critical reading of the manuscript. We thank the cyanobacterial Pasteur Culture Collection (Institut Pasteur, Paris, France), the Roscoff Culture Collection (Station Biologique, Roscoff, France) and the MIT Culture Collection (Massachusetts Institute of Technology, Cambridge, Massachusetts, USA) for providing *Prochlorococcus* and *Synechococcus* strains. We acknowledge the kind collaboration of Instituto Español de Oceanografía for supplying the seawater. We thank Servicio Centralizado de Apoyo a la Investigación (SCAI, Universidad de Córdoba) for DNA sequencing.

### Funding

José Ángel Moreno Cabezuelo and Antonio López-Lozano received a doctoral and post-doctoral grant, respectively, from project P12-BIO-2141 (Proyectos de Excelencia, Consejería de Economía, Innovación, Ciencia y Empleo, Junta de Andalucía, Spain). This work was supported by grants P12-BIO-2141 (Proyectos de Excelencia, Consejería de Economía, Innovación, Ciencia y Empleo, Junta de Andalucía, Spain), BFU-2016-76227-P (Ministerio de Economía y Competitividad, Government of Spain), cofunded by the European Social Fund from the European Union, and Universidad de Córdoba (Programa Propio de Investigación). The funders had no role in study design, data collection and analysis, decision to publish, or preparation of the manuscript.

### Grant Disclosures

The following grant information was disclosed by the authors:
Proyectos de Excelencia, Consejería de Economía, Innovación, Ciencia y Empleo, Junta de Andalucía, Spain: P12-BIO-2141.
Ministerio de Economía y Competitividad, Government of Spain: BFU-2016-76227-P.
European Social Fund from the European Union, and Universidad de Córdoba (Programa Propio de Investigación.
## Competing Interests

The authors declare that they have no competing interests.

## Author Contributions

- José Ángel Moreno-Cabezuelo conceived and designed the experiments, performed the experiments, analyzed the data, prepared figures and/or tables, authored or reviewed drafts of the paper, approved the final draft.
- Antonio López-Lozano conceived and designed the experiments, analyzed the data, authored or reviewed drafts of the paper, approved the final draft.
- Jesús Díez conceived and designed the experiments, analyzed the data, authored or reviewed drafts of the paper, approved the final draft.
- José Manuel García-Fernández conceived and designed the experiments, analyzed the data, prepared figures and/or tables, authored or reviewed drafts of the paper, approved the final draft.

## Data Availability

The raw data are provided in the Supplemental Files.

## Supplemental Information

Supplemental information for this article can be found online at http://dx.doi.org/10.7717/peerj.6248#supplemental-information.

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
