# Peer review of "Differential expression of the glucose transporter gene glcH in response to glucose and light in marine picocyanobacteria"

_PeerJ, doi:10.7717/peerj.6248_

## Round 0.1 · original submission · Major Revisions

Please address all the concerns raised by all the reviewers, specially major comments raised by all the reviewers.

Reviewer 1 ·

Basic reporting

This work reports variability in the expression of the glucose transporter gene glcH in cultures of marine cyanobacteria, in response to increasing glucose concentration and to a period of darkness. The manuscript is overall clear and well-written. The findings are interesting and will be useful to the scientific community. It is however unfortunate that no testing was done in natural environment in order to ground truth findings from culture isolates. I have some concerns about the interpretation of some of the results (detailed below), particularly when using high glucose concentrations in experiments studying a high affinity transporter, and conclusions relating the importance of glucose uptake if transporters expression is down regulated in the dark. Overall the benefits of using sugars uptake for a photosynthetic organism need to be better explained to be more convincing.

The article structure is fine overall, but I found some sentences that should be moved to more appropriate parts of the paper. This would help future readers.
• The first paragraph in the results section really is methods and should be moved where it belongs. Same for L213 to L218 and L241 to L245. All this information relates to the choice of strains studied, glucose concentrations and incubation times and should be written in the method section in a consistent way. I would also suggest improving the justification related to the choice of strains. Please explain why these strains are best suited to interpret what may be happening in the natural environment.
• L271 to 274 is also clearly methods, while L266 to 271 should be in the intro.

The figures could be improved. I don’t find it necessary to use different colors for different bars. This is distracting because the colors mean nothing. Choose one color, preferably black or grey. I also recommend combining results per experiment. The glucose concentration experiment with Prochlorococcus (fig. 1, 2, 3 and 4) should become figure 1, the glucose concentration experiment with Synechococcus (fig 5 and 6) should become figure 2, and the dark experiment (fig 7 and 8) should become figure 3. That would make it easier to compare results for different strains in the same type of experiments. Why are there no error bars for the controls: how that affects statistical analyses?

While sufficient field background/context is provided, the authors self-cite too many times throughout the manuscript and overemphasis their previous work in sentences such as “previous results from our team” (L55), “by our team” (L151, L157), “our team” (L302), “results from our group” (L387); or “we proposed” (L307), “we showed” (L313), “we performed” (L316) while references cite a first author that is not a co-author to this manuscript. In my opinion, scientific publications should not be an advertisement for a research group, even leaders in the field. I therefore recommend removing those mentions to the group or team and simply refer to the publications. Same goes to statements about their ongoing and future work (L384 and L460-462): this should be removed.

L78-80: The authors refer to Muñoz-Marín et al. 2013 but this paper does not demonstrate that the uptake of sugars is beneficial in terms of bioenergetics in comparison to CO2 uptake. This needs to be demonstrated because this statement is core to the work presented here.

Experimental design

The introduction clearly raises important research questions and states how the present work fills an identified knowledge gap.

Overall the experimental design is sound. I have several questions though.
• In the introduction, the authors state that sugars concentration in the ocean are very low: please provide ranges and references. Also explain why the chosen concentrations of 100 and 1000 nM glucose would be relevant to natural ecosystems. This is puzzling considering that the authors make the case that they are studying a high affinity transporter.
• As highlighted above, I am questioning why there was no replication of the control (no error bars in the figures. Or do the error bars for the different treatments consider the variability in the control (i.e. did the authors calculate the treatments standard deviations taking into account the standard deviation in the control).
• It would have been important to test if the addition of glucose to a dark treatment would result in increased expression of glcH. I think it would have been very informative and complimentary to the experiments with glucose additions in the light. Also, L385-387: is it because the cells died under darkness? Or maybe they shut down their metabolism until light is restored? Did you test if glucose uptake is restored after switching cells from dark to light? Indeed, the big question that rises from the authors findings is what is the advantage of using glucose as a source of carbon if the uptake is so low/inhibited in the dark? For a microorganism that has the machinery to do photosynthesis, the ecological advantage to keep the machinery for glucose uptake is unclear if they can benefit from it mostly where/when light is available. I think this is an important point that needs to be better explained in the paper.
• Please justify the choice of 24h of incubation.

Validity of the findings

The results are overall nicely interpreted and put in context. However, the interpretation of the results for incubations in the dark need to be more carefully assessed. For example, in the discussion L344-347, the authors argue that mixotrophy in marine cyanobacteria could be more useful under natural low light conditions. But to me this is contradictory to the result that glucose uptake is reduced under darkness. It is important that the authors improve the discussion related to this hypothesis and their findings.

L227-228: Please be quantitative in this statement. Considering variability in gene expression at 1000 nM for SS120 and MIT9313: is the difference between strains different?

L239-240: Please be quantitative

Paragraph L348-355: Why would glucose concentration be lower in coastal than in open ocean? Please provide data (if not glucose, at least sugars concentration) and references from the literature to support this hypothesis.

L372-376: Interpret the fact that the high affinity glcH is upregulated in response to relatively high glucose concentrations: 100 to 1000 nM are very high for a high affinity transporter that should 1) saturate at relatively low concentrations and 2) never be exposed to such high concentrations in the natural environment.

L411-413: Could this be an indirect effect of glucose addition?

L451: What metabolic needs? If DIC is not limiting and glucose uptake reduced when photosynthesis is reduced, what is the advantage of sugar uptake? Improve explanation related to this important point throughout the manuscript.

Additional comments

L121: growth media for Prochlorococcus and Synechococcus are very different: particularly important is the fact that Prochlorococcus was grown on natural seawater (where was it sampled?) containing sugars while Synechococcus was grown on artificial seawater (no sugars). Can that affect the results?
L136: Why using different absorbance wavelength?
L78: I recommend adding more recent references.
L299: Please define Ks
L330 and L430: Should be “The results presented here”
L333-337: Please remind the readers which strains are LL or HL. That would help non-experts.
L360: Please interpret this observation
L434: Also explain their results with glucose addition because this is relevant to your work.

Reviewer 2 ·

Basic reporting

The manuscript titled ‘Differential expression of the glucose transporter gene glcH in response to glucose and light in marine picocyanobacteria’ by Moreno-Cabezuelo et al, details the differential expression of the glucose transporter gene glcH in response to glucose and light conditions in marine picocyanobacteria (Prochlorococcus and Synechococcus). Authors have analyzed the transcriptional regulation of glcH in varying glucose concentrations and light conditions by qRT-PCR in four Prochlorococcus and two Synechococcus strains. This work is purely an extension of previously reported work by the authors.

As author has complemented the major questions in their earlier publication, I think their data of transcriptional expression profile was a big missing link in their earlier report, which here will be well-complemented and major advancement for the field.

Experimental design

The background and introduction are appropriate and relevant. Sufficient information about the previous findings is presented for readers, which is helping to follow the present study rationale.
The methodology is well executed, and authors had provided details of growth and culture conditions, in which different Prochlorococcus and Synechococcus strains were grown. However, authors didn’t provided detail of the difference in absorbance units of the starting culture for the two sp.

Validity of the findings

The results seem to be valid but not well elaborated. The first paragraph of result section is mostly the repetition of method section. The clades discussed in lines 203, 205 and 206 should be clearly described in the phylogenetic tree, as also reported in Muñoz-Marín et. al. 2017. I would suggest writing ‘Results’ and ‘Discussion’ together.

Lines 70-72 and Lines 90-92, reference should be incorporated.
Line 124, it should be either “as described earlier” or “in El Alaoui et al. 2001”
Lines 229-234, 239-240 and 243-245 consider rephrasing to make the message clear.
Line 298 elaborate Ks for e.g., Ks (Saturation constant)

I have major concern with discussion as it contains a lot of background information, which has already been discussed in introduction. Due to which, the major impact of this manuscript is hidden. For example, authors had discussed about the identification of GlcP and melB/glcH gene in the first paragraph of discussion and in the second paragraph demonstrated the work performed by their group on glcH in earlier studies, which has been done in the introduction too. Thus, the part of discussion that is the repetition of introduction should be omitted which will help to improve the discussion.

Figures can be clubbed together as different part of same figure. By which it would be easy to compare the two profiles. For example, the graphs representing the effect of glucose concentration in different strains of Prochlorococcus, Figures 1-4 should be combined together as Figure 1A, 1B 1C and 1 D. It stands true for Synechocystis sp. i.e., Figure 5 and 6 should be combined as Figure 2A and 2B. Then, Figure 7 and 8 will be Figure 3 and 4, respectively.

Figure legends are poorly written and basically same for each figure. Author need to describe more about result presented in each figure legend.

Additional comments

As authors have presented the qRT-PCR data to investigate the transcriptional expression profile of glcH genes, it would be more convincing if authors would have also checked the regulation at protein level. Comparative western blotting analysis of the proteins corresponding to these genes would be the ultimate data to look upon the physiological regulation of GlcH in the observed conditions.

The manuscript should be rewritten with more focus on the findings of the current study. I have found serious issues with writing of this manuscript, thus the manuscript needs to be well written and well organized prior to be considered for publication.

Annotated reviews are not available for download in order to protect the identity of reviewers who chose to remain anonymous.

Reviewer 3 ·

Basic reporting

no comment

Experimental design

The research question is well defined and performed well.

Validity of the findings

FIndings of research are well discussed with the inclusion of data and conclusions from other peer researchers.

Additional comments

I thank the editor for providing an opportunity to review article “Differential expression of the glucose transporter gene glcH in response to glucose and light in marine picocyanobacteria”.

This article can be accepted for publication after addressing comments below.
1. Interesting work showing the effects of glucose and light on the expression of glcH, however, there is no genetic data to support this conclusion. The author should also test whether or not expressions of other genes are affected and if those changes contribute to bacterial fitness? After taking glucose inside. This will increase the impact of the MS.
2. Please reduce the introduction and discussion part. It’s quite long and unfocused. Discussion part includes mostly about previously published work from the same lab rather than focusing on results in this manuscript.
3. Manuscript figures should be organized in only two figures. As discussed in result section figure 1 should be about glucose uptake and its effect on glcH expression profile across strains. And figure 2 should be the effect of light on glcH expression. There is no meaning of having figures in such a split manner. It creates a distraction. Please consolidate figures.
4. When there is a higher amount of glucose, some strains increase glcH expression, which supports the idea that more transporters could facilitate higher glucose uptake. However, at the same time, transcriptional up-regulation would lead to utilization of more cellular energy. How bacteria make a decision when we think in terms of cellular energy homeostasis? Please discuss.
5. Could you comment on GlcH transporter protein in different bacteria? Uptake ability varies in different strains, is it possible that it is due to protein conformation or sequence differences? Is it possible to do protein alignment for glcH in difference strain to understand similarity and difference across species?
6. Authors also pointed out that possible Hik31 kinase sensing mechanism involved for sugar uptake, when it present in the outer environment, that leads to transcriptional up-regulation of GlcH transporter and higher uptake. Could you elaborate pathway from sensing to transcriptional regulation mechanism?

---

## Round 0.2 · Minor Revisions

Please rewrite/modify figure legends which looks repetitive. Also, please go through Reviewer 2's questions.

Reviewer 1 ·

Basic reporting

The manuscript has improved in clarity, it is better organized and easier to follow. The figures are also much better.

Experimental design

Thanks to the manuscript reorganization, the experimental design is much easier to follow. It is now clear.

Validity of the findings

I have nothing to add in comparison to my previous review. The authors have answered my concerns.

Additional comments

The authors have answered my questions and revised the manuscript accordingly. I am satisfied with their answers and with the current version of the manuscript.

Reviewer 2 ·

Basic reporting

The article “Differential expression of the glucose transporter gene glcH in response to glucose and light in marine picocyanobacterial” by Moreno-Cabezuelo et al. contain interesting and relevant findings for the field.
Introduction section is well executed and is clearly describing the background information as well as the objective for the current study.
Materials & methods section has improved a lot and is providing most of the necessary details in the revised manuscript.
Result section is nicely interpreted and more focused on the current findings in the revised manuscript. However, discussion still is quite long, and authors should work on making their article balanced, yet informative.
Figures are much organized and easy to follow in the revised manuscript which has improved the quality of revised manuscript. However, the language used for the figure legends are still sounding repetitive.

Experimental design

The design of experiments is well plotted and well executed by the authors.

Validity of the findings

The findings of the study are novel. These are clearly explained, discussed and well supported by the previous findings.

Additional comments

Overall the manuscript is providing an important and novel piece of information which will be utilized by the scientific community. Figure legends are repetitive and hence are not acceptable in the current version. Authors need to rewrite the figure legends and should avoid the redundancy in language.

Reviewer 3 ·

Basic reporting

clear and focused

Experimental design

defined question and experiments related.

Validity of the findings

results are discussed well.

Additional comments

I thank the editor for providing an opportunity to review article “Differential expression of the glucose transporter gene glcH in response to glucose and light in marine picocyanobacteria”.
Authors have incorporated and addressed all comments and suggestions. The manuscript is improved and consolidated. I recommend this article to be accepted for publication in PeerJ.

---

## Round 0.3 · accepted · Accept

Authors have incorporated all the changes suggested by the reviewers. I recommend this article to be accepted for publication in PeerJ.

# Reviewer 2 ·

Basic reporting

The revised form of manuscript is much refined and it is now easy to follow. Hence, the manuscript can be considered for publication.

Experimental design

The experimental design is well executed.

Validity of the findings

The findings are clearly explained and discussed.

Additional comments

I recommend this article to be accepted for publication in PeerJ.